# Framing Disaster Risk Perception and Vulnerability in Social Media Communication: A Literature Review

Stefano Morelli [1,2,*] , Veronica Pazzi [2,3] , Olga Nardini [2] and Sara Bonati [4]

1 Department of Pure and Applied Sciences, University of Urbino "Carlo Bo", 61029 Urbino, Italy
2 Department of Earth Sciences, University of Firenze, 50121 Firenze, Italy; veronica.pazzi@unifi.it (V.P.); olga.nardini@unifi.it (O.N.)
3 Department of Mathematics and Geosciences, University of Trieste, 34127 Trieste, Italy
4 Department of History, Archaeology, Geography, Fine and Performing Arts, University of Firenze, 50129 Firenze, Italy; sara.bonati@unifi.it
* Correspondence: stefano.morelli@uniurb.it

**Abstract:** The paper presents the results of a literature review on how social media can impact on disaster risk perception and vulnerability and how these two aspects are interconnected, trying to understand what factors have consequences especially on informational vulnerability. The paper answers to the increasing requests at an international level to move from a technocratic approach to disaster risk management and reduction to a holistic one, where social perspective is integrated. The paper states that this change of paradigm is relevant, especially considering the role that new technologies in communication and information systems are acquiring in disaster risk management and reduction. What emerges from the literature review is that there is a limited scientific production on the topic and further works are desired, to improve knowledge on how new communication and information technologies can impact on vulnerability and risk perception. Furthermore, the two topics are usually discussed separately. However, the role that risk perception can have in increasing or reducing vulnerability deserves to be better discussed.

**Keywords:** trust; communication flow; social vulnerability; risk perception; resilience; DRM

## 1. Introduction

Since the 1980s, the technocratic approach, dominating the Disaster Risk Management (DRM), was accompanied by the seeking to combine exposure (given by the physical components that can be hit by a hazard) to socio-economic and cultural abilities to cope with risk (the social dimension and resilience of individuals and groups) [1]. In particular, first calls for a vulnerability paradigm in disasters came from different scholars engaged in Third World political ecology, natural hazards, and human ecology studies (see e.g., [2–7]). Despite these calls to a social approach, for years the emergency response system has favored the technological and physical sciences, neglecting the contribution that the social sciences could give in the field of Disaster Risk Perception (DRP) and Reduction (DRR) [5–8] sensu UNISDR (https://www.undrr.org/publication/2009-unisdr-terminology-disaster-risk-reduction, accessed on 10 June 2022).

A shift towards the recognition of the importance of a social perspective in DRM has been introduced in the Hyogo Framework 2005–2015 and then in the Sendai Framework for Disaster Risk Reduction 2015–2030, where a socio-centric and bottom-up approach is suggested. Accordingly, the practices for DRM should be based on an understanding of disaster risk also in the social dimensions of vulnerability and people capacity of reaction. This is based on the idea that risks are not only linked to physical phenomena. They reflect relationships with the environment and are culturally, socially, and psychologically constructed [9]. Thus, it is very important to take a multidisciplinary research approach in DRM (see for example [10–13]).

This shift is furthermore important if we consider the widespread diffusion and the increasing role acquired by new technologies, and in particular information and community technologies, in reducing the risk of disasters. The research and application of technological tools for risk management seem to have had a significant impetus, year after year, thanks to the rapid development of social media (in the following SM) technologies and services as new data sources. These tools, in fact, are now part of everyday life and are also prevalent during disasters, because of their capability of sharing information. Thus, they have become key components in contemporary DRM, especially in large-scale and highly impacting events.

After these first experiences, online SM platforms have been increasing their role of support in the processes of DRM and emergency response (among the others see [14]). Recent studies have shown how social platforms can help provide information in real time to first aid and civil protection operators, to better plan response actions and reach as many people as possible. For instance, Ref. [15] states that during Hurricane Harvey people used social media platforms to seek help, overcoming the overloaded 911 systems. Furthermore, SM are increasingly used by authorities as means to provide information to citizens, and at the same time they are one of the methods employed by citizens for voluntary community engagement in the different phases of DRM [16].

Nonetheless, these practices present considerable limits, especially related to the ability to return faithful images of reality if, for example, accessibility problems they pose are not taken into consideration in DRM [17]. Another limit is linked to the challenges connected to fake news sources that often manipulate the information flow during emergencies, frequently using these platforms as trojan horses [18]. Studies on DRP aim to understand how to improve communities' risk awareness using these new tools and how to avoid the spread of fake news.

Considering that the concepts of DRP, vulnerability, and SM belong both to hard and soft disciplines, the purpose of this multidisciplinary work is to explore the interconnections between them, trying to understand how the use of SM impacts on DRP and vulnerability, and what are the factors that affect them. As discussed in the next paragraph, vulnerability is conceptualized in this paper as a dynamic condition acquired over time; thus, a disaster can simultaneously produce experiences of vulnerability and resilience [19–21], including preparedness for new risks. On the other hand, DRP can be defined as the way individuals and groups appropriate, subjectivize, and perceive risks that might or might not be calculated in an objective manner during risk assessments [6,16].

Accordingly, this paper aims to present the results of a literature review on what are the limits and potentialities in using social media in disasters, focusing in particular on how their systematic use may affect DRP and vulnerability. Although disaster risk perception and vulnerability traditionally belong to separate knowledge domains, with few attempts to integrate them [22,23], the authors believe that SM are opening issues about how risk perception and vulnerability are potentially interconnected and how this relationship is also shaping different ways of conceptualizing informational vulnerability. Furthermore, their integration is increasingly desired at the decision-making level, as demonstrated by the calls in framework of the Horizon 2020 (H2020). An example is the call "Human factors, and social, societal, and organizational aspects for disaster-resilient societies" intended to finance projects focused on the capacity of new technologies, media, and tools to raise disaster risk awareness, to improve citizen understanding of risks and to improve functional organization in most fragile and vulnerable environments (https://cordis.europa.eu/programme/id/H2020_SU-DRS01-2018-2019-2020, accessed on 25 June 2022).

It is important to underline that the proposed review cannot be classified as a "systematic review" sensu Denyer and Tranfiel [24]. This review, in fact, does not explore, as an independent research project, a clearly defined question based on existing gaps, but the main purpose is to identify and discuss the existing gaps in literature to suggest further research directions. Nevertheless, research questions that drive the whole process were

formulated at the beginning, and the five steps of the approach defined by [24] have been taken into account. See: Section 2 for the conceptual background from which the analysis starts; Section 3 for the adopted methodology to both select and analyze/classify the papers; Section 4 for a systematization of the analyzed papers; and Section 5 for the discussion on the interconnections and mutual influences among DRP, vulnerability, and SM that lead to propose a framework for DRP and vulnerability in SM communications.

## 2. Conceptual Background

### 2.1. Disaster Risk Perception (DRP)

As mentioned in the introduction, it is very important to take a multidisciplinary research approach to study DRP as the social, cultural, and psychological dimension of risk interpretation [25–27]. In fact, DRP could be directly influenced by people's competences and preparedness [28]. Accordingly, the literature has shown how risk perception has a significant impact on individual and group behavior [29]. The resilience of local communities can be improved if local risk perception is understood [30]. In the same way, risk communication strategies need to have a good knowledge of the local awareness of risks [31]. Thus, studies on DRP aim to understand the interconnections between perception and coping capacity, and to provide communication tools that can improve communities' risk knowledge [32–34].

Despite the many efforts to pinpoint generic reasons for diversity in risk perception, it seems inevitable that individuals and communities base their risk perception on a multitude of factors, including their own experiences [26], memories [35], and, of course, expert risk assessments [36]. It is crucial to understand and acknowledge that risk perception is deeply interwoven within local cultural practices and world views, and that local knowledge on disasters and risks therefore needs to be an integral part in disaster risk management processes. Thus, in this context DRP is seen as the way individuals and groups appropriate, subjectivize, and perceive risks that might or might not be calculated in an objective manner during risk assessments.

### 2.2. Vulnerability

According to the vision of this work, the resilience of local communities can be improved, also, in relation to the real comprehension of vulnerability and to the proper use of its derived practical implication for the DRM. The different disciplines have produced many methods and approaches to understand and measure vulnerability, with limited attempts to integrate them through a holistic perspective. Thus, today it is possible to talk about physical vulnerability, social vulnerability, institutional vulnerability, and economic vulnerability as disjointed notions [35].

Starting from this premise, this paper aims to focus on how the increasing role of SM can shape informational vulnerability. According to the analysis here provided, the specific issue of informational vulnerability takes on particular relevance. It deals with the access to availability and understandability of information within affected communities. In general terms, the lack of informational resources, digital disparities, or difficulties to access information in the SM context affects the capabilities of people who are dependent on these sources of information to deal with disaster risks with consequences on their level of exposure, susceptibility, and resilience capacity. Furthermore, informational vulnerability may interact with other socio-structural conditions, particularly geographical location. For instance, the informational vulnerability among rural people in some part of the world is further augmented by the low literacy levels and lack of relevant technological skills necessary to enable the learning and processing of information. Consequently, an underrepresentation of the population may ensue, e.g., during emergencies (see [17,37–44]).

Furthermore, this paper aims to investigate vulnerability as a dynamic property. Vulnerability has been conceptualized as a dynamic concept in social sciences and today on this there is a high agreement in the scientific community. The dynamic property of vulnerability is based on the recognition that vulnerability depends on a multiplicity of

different features that may change over times and places, overcoming the idea that it is a specific attribute of some individuals (see among the others [45–47]). These features can be synopsized in the concept of diversity, which should not be interpreted exclusively as the sum of different characteristics, methods, and contexts, but also as the capacity resulting from situations usually perceived as disadvantageous [19,48]. Thus, these two concepts (diversity and contexts) are integrated and coexisting in different situations. This is in line with the idea of diversity in a social perspective, both as a component of vulnerability which places some individuals at higher risk, and as a beneficial component that contributes to people's coping capacity [49].

### 2.3. Scope of the Literature Review: Integrating Disaster Risk Perception and Vulnerability and Identifying Gaps

As mentioned in the introduction, this paper aims to investigate how social media can impact on DRP and vulnerability, in particular, on informational vulnerability, and also how DRP and vulnerability are interconnected. In particular, it aims to investigate how the increasing use of social media as a communication tool in disasters is shaping or may shape the way DRP and vulnerability interact and are conceptualized. Because this paper is not field-research based, it does not pretend to be able to provide a final answer, but aims to open a discussion and to identify potential ways to investigate the topic in the future. Thus, the starting point is how the two concepts have been discussed till now in literature and what are the open gaps. This examination adheres to the following questions: can SM improve disaster risk perception and reduce vulnerability? Can SM also be responsible for producing the opposite? What are the factors that have a main role in this?

## 3. Methodology

This section describes the qualitative method adopted for building the conceptual framework (presented and discussed in Section 5) to frame DRP and vulnerability in SM communication. In particular, given the multidisciplinary aspect of the topic, some of the steps of the procedure proposed by Jabareen 2009 [50] have been adopted. Thus, after the identification of the texts (the following Section 3.1 describes how they have been selected), the phases of identifying and naming concepts have been carried out. The results, i.e., the four main concepts identified, are summarized in Section 3.2. Furthermore, the phase of deconstructing and categorizing the concepts identified and that of integrating them, have allowed to derive some secondary concepts. These have been employed to design the conceptual framework and to identify the mutual relations among SM, DRP, and vulnerability.

### 3.1. Data Sources

The traditional literature review (sensu [24]) proposed in this paper has its origin in the work carried out to build the knowledge based domains presented in the project deliverables 2.1 [49] and 2.2 [51] of the EU LINKS project ("LINKS-Strengthening links between technologies and society for European disaster resilience", funded under the H2020 call [52]), but it has been implemented as described below. It entailed the analysis of worldwide scientific documents, taking into consideration conceptual, theoretical, and empirical works which describe vulnerability and DRP in relation to the use of SM in the different phases of the Disaster Management Cycle (DMC).

Data selection and collection were based on texts as secondary data sources. No other qualitative sources, such as interviews or surveys or primary data coming from datasets, were collected and considered. Papers/works have been identified, in accordance with Vom Brocke et al. 2015 [53], by searching on Scopus, Web of Science (WOS), and Google Scholar websites. The workflow is schematized in Figure 1.

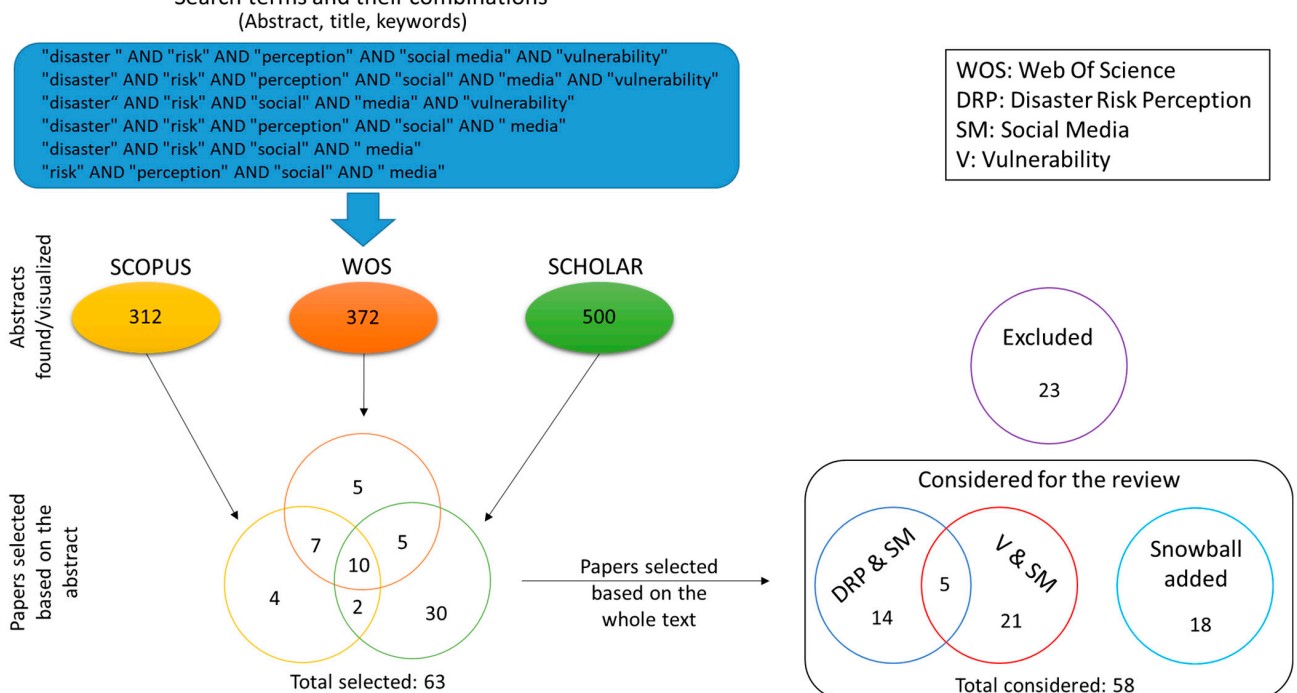

**Figure 1.** The workflow of the procedure applied to search and select papers.

First of all, to select papers on the basis of their titles, abstracts, and keywords, the search terms and combinations shown in the blue rectangle in Figure 1 were applied. They are:

[disaster] AND [risk] AND [perception] AND [social] AND [media] AND [vulnerability]; OR;

[disaster] AND [risk] AND [perception] AND [social media] AND [vulnerability]; OR;

[disaster] AND [risk] AND [social] AND [media] AND [vulnerability]; OR;

[disaster] AND [risk] AND [perception] AND [social] AND [media]; OR;

[disaster] AND [risk] AND [social] AND [media]; OR;

[risk] AND [perception] AND [social] AND [media].

The search allowed to identify/visualize/collect 312 works in Scopus, 372 in WOS, and 500 in Scholar. Among them, based on the reading of the abstracts (second step of the procedure) only 23 works from Scopus, 27 from WOS, and 47 from Scholar, for a total of 63 works, were selected. In particular, 10 works have been identified in all database platforms, 7 both in Scopus and WOS, 5 both in WOS and Scholar, and 2 both in Scholar and Scopus.

The third phase of the process implied the reading of the whole text of each selected work. This phase allowed to keep 40 papers, to exclude 23 works, and to add 18 more articles based on a snowball procedure. Among the 40 works selected during the second phase, 5 of them explicitly discuss DRP, vulnerability, and SM, while 14 focus on SM and DRP, and 21 on SM and vulnerability. The 23 works have been excluded according to the following criteria: they do not consider informational or social vulnerability, or they do not consider the role of SM in relation to DRP and/or vulnerability.

### 3.2. Concepts Identification

Before carrying out a unique analysis procedure for the vulnerability and DRP works, a preliminary schematization (some basic information such as the disasters analyzed, the SM employed, and the involved citizenry) was prepared. It allows to identify the most relevant features/concepts for understanding both vulnerability and DRP in relation to the use of SM in disasters. The approach for identifying variables was limited to the concepts that support one another, articulating their respective phenomena and establishing a framework-specific philosophy [49–51]. Thus, they are ascribable to four broad categories as well codified in the pertinent literature [26,54]: trust, social aspects, individual aspects, and information/communication flows. These categories are briefly summarized here below.

#### 3.2.1. Trust

The role of this feature is focused on the trust of the quality and content of information exchanged via SM, which are exploited as official communication means based on direct interpersonal relationships (i.e., the use of formal network from authorities in charge to beneficiaries of messages in primary or derived connection) versus the role of more traditional sources and outlets of information [14,28,55–57]. The problem is also inherent to the source of information because of widespread public criticism in credibility and responsibility from the population [42,57,58]. This is an issue that can recur both in ordinary and extraordinary moments of risk management. However, in the analysis of informational vulnerability, the vision of the trust relevance can be even overturned during emergencies [59,60]. The matter arises in the use, or abuse, of communication networks by people who suffer from a critical situation (in progress or imminent) when they turn to the authorities for requesting rescue. In this case, trust has a reciprocal value that goes beyond the opportunistic or cultural reasoning from one side and dogmatic and rigid technocratic formulations on the other.

#### 3.2.2. Social Aspects

The social aspects explore the attitudes towards SM of different groupings identifiable in a geographic area (countries, urban/rural regional, wide societies, little communities, and so on) but also those processes and conditions, such as marginalization and social disparities, influencing their current and likely future use in the emergency management cycle [37,42,58,61,62]. It includes the attitude of a connected network of people towards the disparities, vulnerabilities, inclusivity, and social support as motivators of disaster preparedness, as well as an overall interrelation of a populated area for the adequate preparedness to management of risks [63–66]. This implies, in many cases, government-civil society relations, but also among different groups in which cultural barriers have been consolidated over time.

#### 3.2.3. Individual Aspects

The individual aspects include all those elements that influence people's personal capacity to prepare for and respond to disasters. They consider the individual features of citizens including their awareness, level of knowledge, behavior, emotional status, and experience [67,68]. Furthermore, the individual aspect takes into account expectations towards authorities, individual, socio-structural, and situational vulnerability perception that shape how people understand and act, based on perception about hazards [16,17,68].

#### 3.2.4. Information/Communication Flow

This item is focused on aspects that are directly or indirectly connected to the retrieval, production, dissemination, and exchange of information at all social levels. Among them it considers multiple concepts on the type, reliability, speed of information flow from authorities, and the way to communicate (or re-spread and share) to end-user effective messages according to different hazard scenarios, and social effects in mass emergency situations, from the preparation phase to last reaction [55,67–69]. Therefore, this category includes the concept of acquisition of critical and useful information in the limitations in resources and

workforce both for specific vulnerability evaluations and awareness, and risk perception strengthening. In this perspective, it places the centrality on the citizen involvement in different phases of disaster management thanks to the growing technological support of SM for a systematic information transfer and positive effects production [37,42,55,70]. This category is completed by the aspects that include the circulation of unofficial information from peer to peer, superimposed to those of official and formally reliable channels for dispersion of information or their concrete lack in some crucial moments.

## 4. Results

From the analyzed 58 articles some basic information was extracted. This information is summarized as an example in Table 1 for the 5 works focused on DRP, vulnerability, and SM, while it is listed in Tables A1 and A2 for all the other analyzed works. The following common aspects are included in these tables: kind of occurred hazard, exploited technological services, investigated stakeholders, and primary and secondary application filed in relation to DRP and vulnerability.

The natural hazards column includes hazardous natural phenomena that have a negative effect on humans. In this summary weather-related hazards and exogenous/endogenous geological phenomena are specified according to their official classification in their respective geophysical, meteorological, hydrological, and climatological fields (when deductible in the manuscript). In these cases, one or more of them is described.

The technological services list social networking services and IT communication applications used for exchanging news (in microblogging, portals, and sharing platforms), and they highlight where participatory online activities have been used in relation to the hazards described in the first column. However, when these are not made explicit in the text, the macro-category to which they belong is reported. The list of stakeholders includes those who have been beneficiaries, managers, or in any case involved, in various capacities, in the use of the previously recognized web services.

The last two columns describe the main thematic spheres of the social use that dominates the action and/or reaction of stakeholders in relation to the events. The primary fields have been described in Section 3.2, and they have been used as entry points to guide the process of analysis. The social one is mainly connected to vulnerability, individual aspects, and trust derived by disaster risk perception, while information/communication flow is an independent field to consider that could interact with both of them. On the basis of these four macro-fields, the authors went to analyze the papers in detail, observing what were the main topics that fell into each of these fields and how they interacted with each other. Thus, the secondary fields emerged as the main topics identified in the analyzed papers, as key concepts that interact with DRP and vulnerability, especially the informational ones. These fields can be considered sub-fields of the previous ones and in particular:

-    Social aspects: social, demographic, and geographic differences, and accessibility;
-    Individual aspects: awareness and experience;
-    Information/communication flow: quality of information and reliability;
-    Trust: trust.

The secondary fields will be introduced and discussed in Section 5 since they were not postulated at the beginning but were as a result of the analysis carried out.

The 98.28% (i.e., 57 over 58) of the identified papers were published in international sector journals and just 1 work is a conference proceeding. It shows a high scientific attention starting from about ten years after the birth of the first SM platforms (2003–2004). This time span includes the period in which technologies (PC, tablet, and smartphones) and related hardware platforms have implemented and developed free internet softwares related to SM practices/services, which are still conceptually valid (and in most cases still active even if in the most up-to-date version).

**Table 1.** List of papers considered in this review work and focused on DRP and vulnerability. For each work (column 1) are highlighted: the considered hazard/s (column 2), the used/analyzed technological service/s (column 3), the involved stakeholders (column 4), the main field, among those conceptualized in Section 3.2, in which the use of SM primary emerges (column 5), and the second field, among those introduced in Section 5.

| Analyzed Papers | Kind of Hazard | Technological Services | Stakeholders | Primary Concept/s in Relation to DRP and vulnerability | Secondary Concept/s in Relation to DRP and Vulnerability |
|---|---|---|---|---|---|
| Lai et al., 2018 [37] | Cyclones/typhoons, floods (multi-hazard) | Mobile technologies | Urban and rural inhabitants | - Social aspects<br>- Individual aspects | - Accessibility<br>- Social, demographic, and geographic differences |
| Tauzer et al., 2019 [71] | Floods | Social media (Twitter, WhatsApp, and Facebook) | Community members | - Social aspects<br>- Information/communication flow | - Awareness<br>- Accessibility |
| Yue et al., 2019 [72] | Wildfire | Social media (Twitter) | Geo-tagged data | - Social aspects | - Social, demographic, and geographic differences |
| Hansson et al., 2020 [42] | General overview (flood, fires, tsunamis, earthquakes, and hurricanes) | Social media (Examples of Facebook, YouTube, Twitter, Instagram, and WhatsApp) | Review-based analysis | - Social aspects<br>- Information/communication flow | - Quality of information and reliability<br>- Accessibility<br>- Social, demographic, and geographical differences |
| Dargin et al., 2021 [57] | Hurricanes | Social media (Facebook, Twitter, and Nextdoor) | Population groups in the aftermath of three major U.S. hurricanes occurring between 2017 and 2018 (Harvey, Florence, and Michael) | - Social aspects<br>- Trust<br>- Information/communication flow | - Trust<br>- Quality of information and reliability<br>- Social, demographic, and geographic differences<br>- Awareness |

The growth in productivity has been constant over time. However, the issue of vulnerability has currently had a greater increase with a peak of seven articles published in 2019 and DRP has a peak of five articles published in 2021 (Figure 2). Nevertheless, it is not possible to link these peaks with a particular and evident situation (e.g., the COVID-19 pandemic) and it is out of the scope of the present work to carry out this kind of analysis. From Figure 3 it is also evident that the highest production (or indeed the most studied and analyzed areas) is in North America for vulnerability and Asia for DRP. For some papers (i.e., the reviews) it is not possible to define a study area, so they have been grouped under the "N/A" category (green in Figure 3), that means "not applicable".

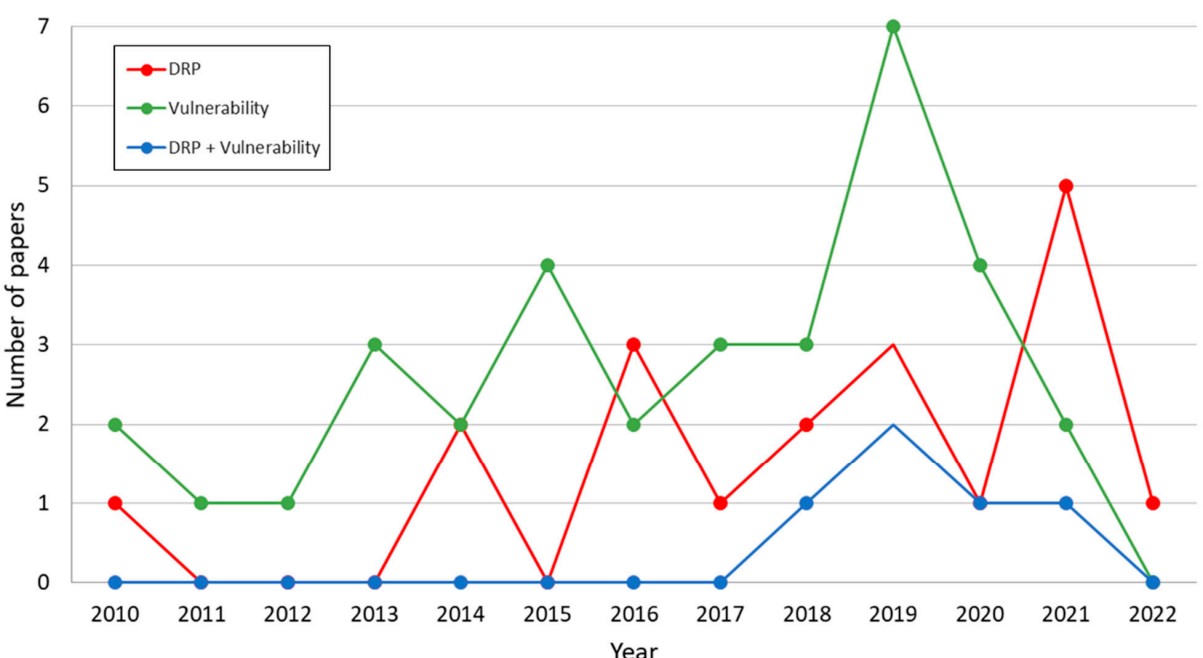

**Figure 2.** Distribution of the papers production over the time (years) about DRP and SM (in red), vulnerability and SM (in green), and about DRP, vulnerability, and SM (in blue). The number of papers published in the 2022 is referred to the period of January–March.

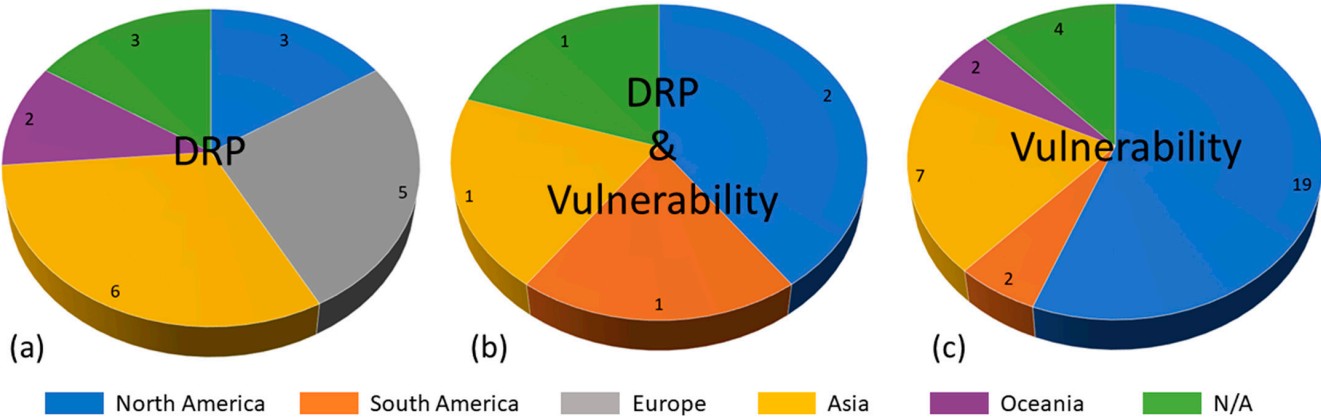

**Figure 3.** The spatial distribution of the (**a**) DRP, (**b**) DRP and vulnerability, and (**c**) vulnerability analyzed papers. The "N/A" category groups those works where it is not possible to define a specific study area (i.e., in the review works).

Most of the analyzed papers have a multi-hazard approach (Figure 4a) as well as, among those that are focused on a single hazard, hurricane is the most considered, especially in vulnerability studies. The first articles to be published do not delve into the details of the

single technologies, but are more generic in the analyses, although some of them were born some years before the time range analyzed. These include the most globally widespread platforms (e.g., Twitter in 2006, Facebook in 2004, YouTube in 2005, WhatsApp in 2009, Sina Weibo in 2009, etc.), but they are also those they took the most time, as a novelty, to find a space in the social context of the people life and a direct role in the safety communication. Figure 4b shows the distribution of the SM analyzed: Twitter and Facebook are the most employed both for vulnerability and DRP.

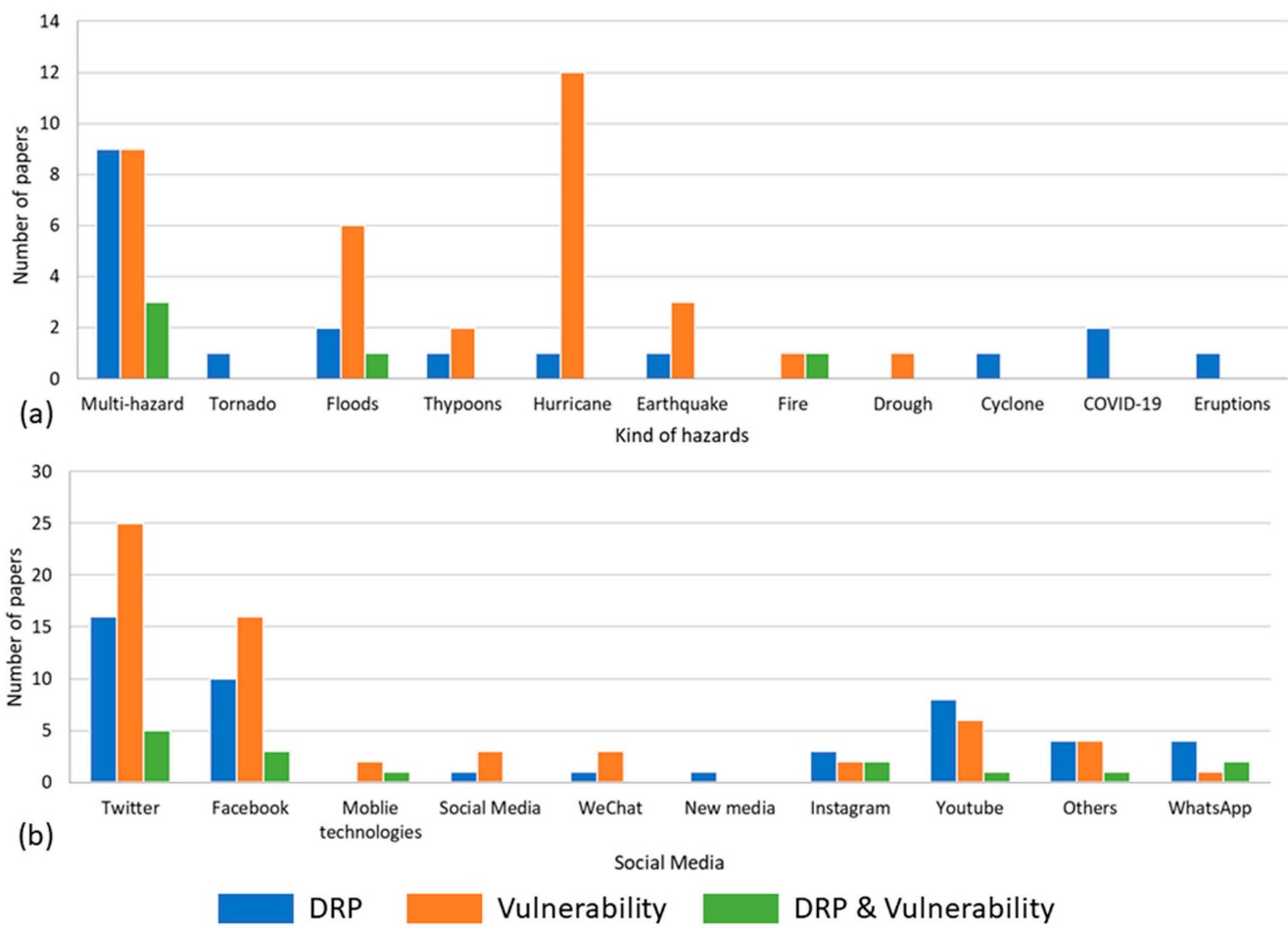

**Figure 4.** (**a**) Kind of hazards and (**b**) SM analyzed in the papers that focus on DRP (in blue), on vulnerability (in orange), and on DRP and Vulnerability in green. The class "others" in panel (**b**) refers to technologies not specified as mobile technologies or information technologies.

As shown in Figure 5, the most used SM services in the DRP and vulnerability fields are Twitter, Facebook, YouTube, and Instagram. Each one is used independently from the others even if, in some cases, they were activated under the same hazard conditions (in general terms, the stakeholder typology is the one who changes). Twitter was more useful in understanding the field of vulnerability while Facebook was successfully used for the study of DRP. The dimension of social aspects is more represented than the others. Secondly, the information/communication flows. Then the individual aspects follow, and ultimately the trust emerges. WhatsApp and other mobile technologies and services still occupy a narrow niche for the topics here considered. In this second minority group, but emerging according to the publication trend, all application dimensions are represented as regards the DRP.

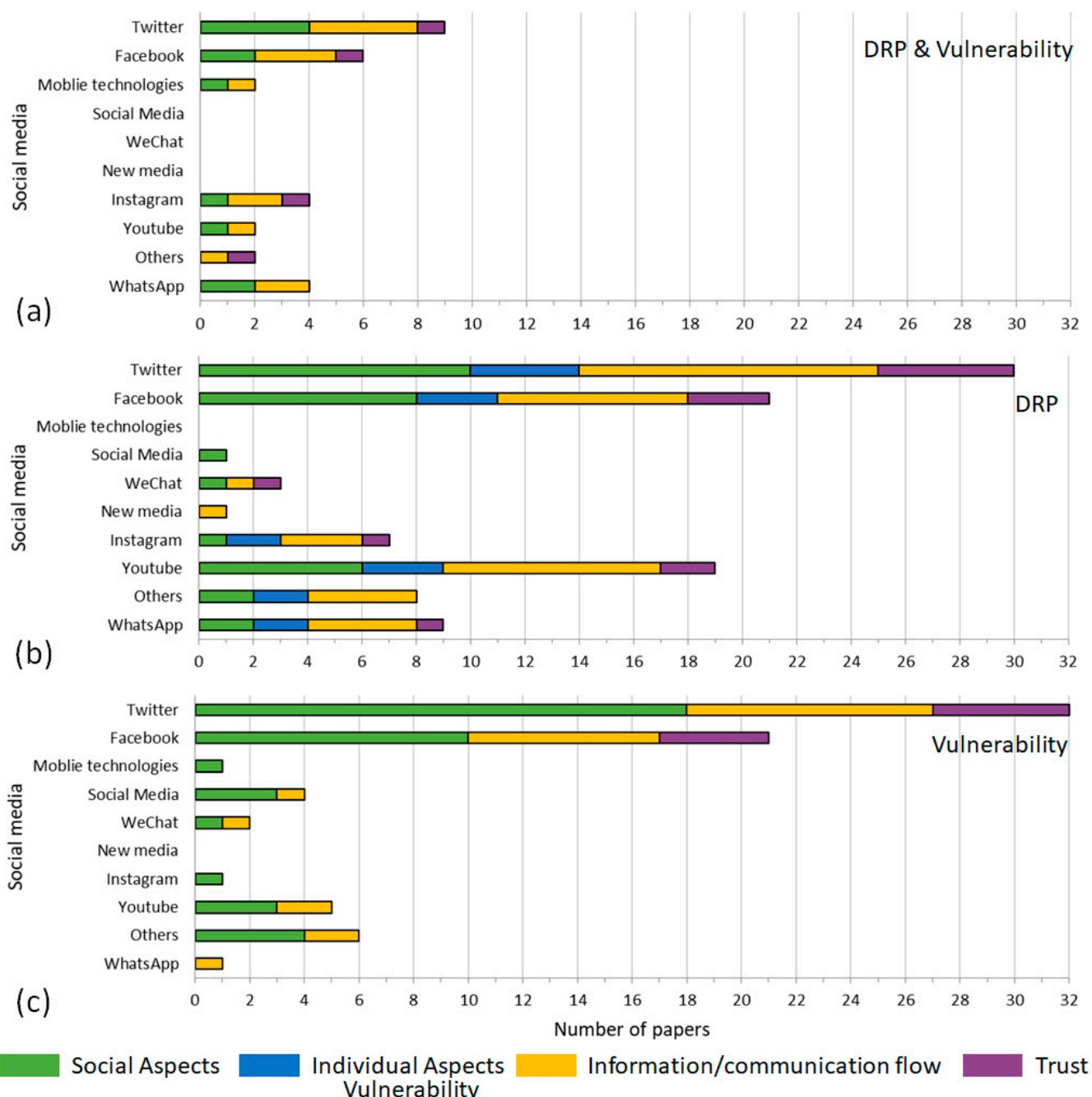

**Figure 5.** Use of the different SM in relation to the four main aspects identified (light green for social aspects, blue for individual aspects, yellow for information/communication flows, and purple for trust) for (**a**) DRP and vulnerability, (**b**) DRP, and (**c**) vulnerability studies.

## 5. Discussion

According to the literature review here provided, what emerges is that SM could have an important role especially during disasters in:

- Increasing situational awareness (checking information) (see, among the others, [57,67,68,70–74]);
- Disseminating alerts (see, among the others, [71,75,76]);
- Sharing information (see, among the others, [37,42,57,63,67,70,76–78]);
- Mapping crisis and optimizing decision (see, among the others, [64,79]);

- Coordinating rescue operations, such as coordination and monitoring of evacuation procedures (see, among the others, [64,65,75,79–85]);
- Establishing (spontaneous) volunteer organization (see, among the others, [57,86–88]);
- Connecting citizens themselves during or after disasters, not only to start volunteer efforts and share information, but also to obtain mutual psychological support and reduce worries, such as to reduce social and spatial distances and digital disparities (see, among the others, [69,89]);
- Understanding the level of awareness/risk perception of the people (e.g., [55,90], their reaction [75], and behavior [40,66]).

According to Chatfield et al. 2014 [76], SM can be a tool to reduce the lack of citizen-centric DR communication; furthermore, Sarker et al. 2020 [91] say that the potential of big data strategies can help mitigate the risks and impact of socio-ecological vulnerability. However, Velev and Zlateva 2012 [63] state that social media cannot and should not supersede current approaches to disaster management communication or replace existing infrastructure, but if strategically managed, they can be used to bolster current systems.

Although all these examples are useful to understand the potentials of SM initiatives, most of the authors agree that the discussion on the role of SM in disaster risk management is still open, e.g., thinking on the difficulties in coordinating spontaneous actions with official systems of response or the risks of disinformation and fake data.

Accordingly, the purpose of this section is not to discuss about the usefulness and potentials of SM in DRM, as it has been already widely discussed in literature (e.g., [57]), but to identify what are the main challenges or factors that impact on the use of SM according to the scientific papers taken in exam, particularly the implications they have on vulnerability and risk perception. Although some works have already tried to do this, most of them have not considered vulnerability and risk perception literature together, thus not considering the implications that SM can have on both.

The results of this proposed literature review has shown that the four primary application fields associated to DRP and vulnerability are mainly challenged by accessibility (access to information), quality information, reliability, trust, awareness, experience, and social, geographical, and demographic factors. These features (the secondary concept/s in Tables 1, A1 and A2) are all taken in exam at follow to understand their implication on risk perception and vulnerability, especially informational one, that emerged as one of the most considered dimensions of vulnerability taken into account in the papers analyzed. Thus, Figure 6 summarizes the results of this analysis, i.e., the interactions between disaster risk perception variables (e.g., awareness, knowledge, and trust), vulnerability ones (e.g., accessibility, reliability, and socio-geo-demographic factors), and the information quality provided by SM. The following discussion aims to clarify this interdependence, therefore answering the starting question of the paper, that aims to understand how social media impact on DRP and vulnerability, and how the two concepts are discussed together, eventually unraveling what are their interconnections that can help to understand the limits of a disaster communication provided through SM, as represented in Figure 6.

### 5.1. Accessibility

Accessibility is one of the main topics raised by the considered literature in relation to vulnerability but also DRP, especially considering how accessibility to resources has consequences in producing informational vulnerability and in affecting quality information. In disaster vulnerability literature, accessibility can be defined as the ability to use the available resources that ensure livability, which depends on the socio-economic relations established in a society [92,93]. So, accessibility is, first of all, connected to the kind of power relations existing at local and global scale. Focusing on social media, accessibility depends, in particular, on the availability of resources that give people the possibility to be connected/online and to connect with others. This enables them to be identified, recognized, and considered into the relief system and to receive information.

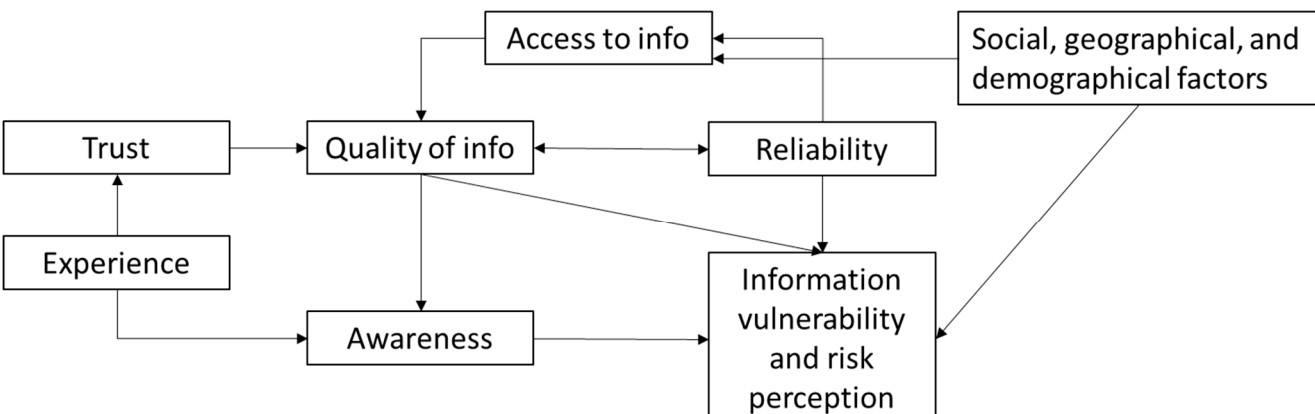

**Figure 6.** The mutual connections among the factors that shape and change vulnerability and DRP in relation to the informational vulnerability use of SM.

According to the scientific literature analyzed, accessibility is mainly discussed through a citizens-perspective. The main points that emerge on accessibility are about:

- Material accessibility: it refers to access to material goods that help to satisfy basic needs and to be able to respond to disasters (e.g., [42]). Talking about SM communication, it refers in particular to the availability of technological devices and internet connection (see [37,41,84,94]);
- Physical and sensory accessibility: it refers to the physical and/or sensorial (in)ability to use specific platforms or communication systems (see in particular [43]);
- Cultural accessibility: It refers to the access to information and knowledge, as well as to education. It could have an effect, in particular, on how people respond (thus, their awareness and behavior) (see [37,42]). The geographical context could also affect the capacity to access information conditioning the way people use social media (see also implications on informational vulnerability [37,85,94]);
- Relief accessibility: It refers to the possibility of access to the relief system, i.e., sending requests and receiving support (e.g., [42,94]). This point especially refers to the possibility people have to highlight their needs and see them answered by disaster management organizations. This point is strictly dependent on the above ones.

In particular, social and spatial disparities are considered as limiting the access to technologies, and therefore to information (especially quality information). They are also at the basis of the inability of marginalized people to see them represented in disaster risk management (see e.g., minorities, homeless, refugees, etc.) [39,40]). As Nicholson et al. 2019 [17] state, using SM platforms as sources could increase the risk that vulnerable people are under-represented in data, and, consequently, that socially vulnerable groups could be left behind during disasters.

Furthermore, Howard et al. 2017 [59] state that the very different and specific information access choices made by certain population groups are of particular interest, and they implicate that that group would not receive the information, if it is provided via a medium not usually used by this group. The problem is also how people decide to use SM and if they use them to be informed [77].

*5.2. Quality of Information and Reliability*

Quality of information is a core topic in the discussions on the potentials of SM in DRM. The access to quality information can increase DRP and trust in government (see [70,74,95]). It is also strictly connected with the reliability of information. Some works show the role SM platforms can play (e.g., [57]) in leading a better decision, as in Weyrich et al. 2021 [96], especially if information is provided by who is at least confident with the problem (e.g., in their work people trained in meteorology). However, the access to quality information,

such as reliability, could be affected by the high amount of information that circulates and the quantity of people that participate in its circulation, with the risk of not having enough control on information. The main challenges associated with using these platforms to receive information, whether inadvertently or deliberately, are the redundancy or inconsistency of information and the dissemination of rumors, with the risk of their propagation but also manipulation or no volunteer alteration of the messages [16,55,62,67,69,77].

Another relevant point is about the level of detail provided especially in SM. The space in SM is usually limited, such as the level of "attention" of the people visiting them [68,69]. This has virtually consequences on the quality of information and the ability of people to check it and identify potential misinformation/disinformation. The point is that citizens share more than emergency management organizations, with higher risk of dissemination of false or inaccurate information which can also jeopardize their safety [77]. According to Silver and Matthews 2016 [69] SM are based on systems of control by the other participants that should ensure the reduction of rumors. However, this could not be enough, especially if people are not aware how to distinguish between good and bad or fake information. Imran et al. 2015 [83] and Mohanty et al. 2021 [97] have underlined that a solution could be the adoption of automatic/data learned models to draw and filter information.

Furthermore, reliability refers also to the reliability of the communication system in providing information everywhere and when needed [60]; not all people and places enjoy the same level of reliability in receiving online information. Thus, reliability affects accessibility to quality information. The different geographical and social distribution of reliability is usually a consequence of social and demographic differences, such as social inequality [42,57]. About demographic differences, Reuter et al. 2019 [58] have analyzed the citizens' perception of social media use in emergencies across Europe, identifying that attitudes and behaviors are significantly related to gender, regardless of the national and cultural context, which can give an understanding of the level of perception. Across all countries and respective risk cultures, it was seen that women are significantly more likely to use SM in a disaster than men, wherever they live. So, there is a significant relationship between downloading an app and gender.

To conclude, Moorthy et al. 2018 [60] state that reliability is an essential aspect of disaster management communication systems, as these systems may be deployed in remote regions, and at times not accessible by the usual communication modes, especially in areas such as deep oceans and mountains (see about Section 5.6).

*5.3. Trust*

The main concepts relating to trust remain firmly anchored to the descriptive characteristics that define the macro category mentioned in Section 3.2.1: quality and content of the exchanged information, source of information, and its relevance and credibility. All these issues are perfectly focused on DRP and vulnerability both in ordinary and extraordinary situations of risk management. Therefore, within these boundaries it is still possible to formulate a reasoning.

As Mehta et al. 2017 [56] discussed, there are four different levels of relationship that determine the use of SM as tools of communication in disaster response:

- Trust that emergency management organizations have in the information provided through such platforms and in the platforms themselves;
- Trust that citizens that use such platforms have in the emergency management staff, governmental structures, etc.;
- Trust between governmental and non-governmental organizations;
- Trust that the personnel working in the emergency management organizations have in the system they work.

They can be defined as different levels of "connection" among different stakeholders that are involved in the information and communication flow. In this perspective, SM are potential tools of social connectivity that can work to support the disaster risk management system. At this analysis, another important layer should be added: i.e., trust that citizens

have in the information provided by other citizens that are not part of the emergency response staff.

As anticipated, trust is mainly dependent on who provides information (higher trust in experts, for example, see [96]), who shares the information, and how information is collected and used for the different purposes (for example a wrong use of data provided by someone could challenge his/her trust in the social media). Thus, trust has to be considered an important point to make SM a useful tool in DRM. In a positive perspective, trust in SM is connected to awareness. This means that if one trusts in the communication system, he/she could receive benefit, obtaining useful information that could have consequences on his/her level of awareness. Moreover, if one knows how the system works, he/she could have higher trust and be able to identify reliable information. Last case, trust is dependent on experience, as this one has consequences on the level of trust in the information widespread [55].

Thus, today trust in information systems and information providers is one of the main reasons why SM are not used in emergencies both to send and receive information [56]. McCallum et al. 2016 [14] and Veer et al. 2016 [65] state that the role of SM in disasters seems to be influenced mainly by the trust that citizens have in the authorities in charge of risk prevention and vulnerability management, as well as in applying SM as a source of information during an emergency. On the other hand, according to Hassan et al. 2022 [98], risk perception has a significant effect on trust in government and self-efficacy. This means that people who perceive the risk are likely to increase their trust in the government during a public crisis. Positive relationship between risk perception and trust in government would be stronger with higher social media usage to acquire information.

To conclude, about mistrust in authorities, Geng et al. 2021 [68] highlight how users who do not trust authorities' response to disasters, could question the credibility of the media. Similarly, Cornia et al. 2016 [28] have shown that trust in the provided information is strictly connected with how different communities perceive the level of political influence exercised on the different media.

*5.4. Awareness*

Awareness is linked to who can share and participate in the communication flow. Thus, it is particularly influenced by the level of accessibility to the system. SM could influence timely situation awareness (among the others [67]). However, it depends on how often information is spread. Situational awareness has been defined in literature as the perception of environmental elements and events with respect to time or space, the comprehension of their meaning, and the projection of their future status [99]. Therefore, situational awareness is embedded within a cognitive model of human activity in a dynamic system and is influenced by task factors and individual factors [100].

Some authors state that the general public's risk awareness is relatively poor and is greatly affected by the attention degree [68], and that situational awareness can be especially influenced when information providers are people responding to the event (see [61]). Timely dissemination of precise and comprehensible disaster warning to populations at risk and to relevant disaster authorities, may minimize loss and damage.

Other factors that could influence awareness are frequency of information, level of attention of the users, and information sources. About the frequency of information, that is also connected to the level of attention, what emerges is that it is not only important to receive information on risks, but also to have a reminder of them from time to time, otherwise people tend to forget and lower their risk estimates (see [101]; see also [60,102]). Thus, the more time elapsed from the last event, the lower people's awareness [101,102]. Thanks to the possibility of "re-posting", SM can play an important role in sharing information over time. However, as in Moorthy et al. 2018 [60], people use these platforms to send and receive information on a variety of aspects regarding the disaster. This could impact on the quality of information producing rumors, as discussed above.

Another important point raised by Geng et al. 2021 [68] refers to the "geographical distance" and the impact this factor can have on awareness. It means that people far from the event are less active on SM, during the event, compared to those nearer to it. This implies that even if SM users understand the severity of disasters, they have a low-risk awareness.

About information sources, Kaufhold et al. 2019 [70] state that knowing the social media accounts of local and national emergency services or following their information on how to prevent and stay safe during a disaster could improve people's DRP and consequently awareness. Nevertheless, only about half of the interviewed people thought it essential to look for and download apps released by the emergency services to stay informed during an emergency, or to read what to expect from the emergency services' social media.

To conclude, Tauzer et al. 2019 [71] say in their work that SM could provide voice to marginalized communities as well as a mechanism to raise local awareness. Nevertheless, one of the main critical issues emerged from the analysis carried out in this work is that SM could increase the situational awareness in some groups but at the same time reduce it in others, with potential increase of social disparities.

### 5.5. Experience

Experience has been represented in Figure 6 as a factor that can directly affect trust and awareness, and indirectly quality of information/reliability. Furthermore, the experience of past event/s can influence how SM are used, with potential consequences on vulnerability and risk perception [16,17,59,60,65,68,98]. Accordingly, people with previous experience of disasters could have two different approaches to the use of SM in crisis communication:

-   First, they could use SM both to obtain and provide information. According to Cheng et al. 2016 [103], people with previous experience that use SM to obtain information about a new emergency, feel less anxious about the future. This is also because experience can influence people's capacity of understanding messages about what is happening [42,59,95]. About the second use, direct information provided by people with previous experience better resonates with those experiencing the same (or similar) situation and can influence the personal risk judgements, and thus the information shared [103–105]. Another aspect that emerges is that who has experienced an emergency knows text messaging work better than voice [89]. Finally, the availability of information can generate empathy for others going through the similar experience [73];
-   Second, people with previous experience of disasters are willing to trust more in their personal experience than in the information provided by SM channels of communication [42]. Therefore, they tend to ignore SM information about hazards, considering it inaccurate (about trust and experience see also [42,55,57,103]). Furthermore, those strongly affected by a disaster may not be inclined to post anything on SM [73].

Both these approaches can have consequences on the levels of awareness and consequently of informational vulnerability and disaster risk perception. For example, previous experiences can either limit people in adequately acting after having received information about a hazard or a disaster [42], or enhance their sense of risk [37,54,59,71,73], also according to the self-regulation theory [42].

To conclude, there is another level of experience to be considered in relation to the use of SM in disasters. This is about the indirect experience of the disaster. A quite common phenomenon is that of experiencing an event via SM and of carrying out post-disaster actions even not in the disaster area [54,103].

### 5.6. Social, Demographic, and Geographical Differences

Social, demographic, and geographical differences emerge from the analysis of all the factors described above. They represent a constant variable that could condition significant disaster risk perception, producing informational vulnerability.

For a lot of authors, the different use of SM reflects the existing social and structural inequalities [37,40,54,55,58,62,69,85]. In particular, Wang et al. 2019 [39] discusses the differences between physically and socially vulnerable groups in opening and participating in disaster-related SM conversations. Physically vulnerable groups were identified as more likely to use SM with the aim of reducing their susceptibility, engaging more in conversations on, e.g., preparedness and situational updates. Moreover, according to Lai et al. 2018 [37], the diffusion and use of smartphones, coupled with demographic and geographical differences, reflect disparities in disaster information behavior and preparedness.

Other authors, like Zou et al. 2018 [40], have studied the social and geographical disparities in the use of SM to verify if these two disparities could produce irregular responses in case of disaster and affect community resilience. Results by Zou et al. 2018 [40] show that social-demographic conditions affect the disaster-related SM use during preparedness, response, and recovery. However, disparities emerged mainly in the response phase.

Similarly, the analyses of Lai et al. 2018 [37] and Hansson et al. 2020 [42] show that there is a correlation between demographic and geographical differences and informational disparities. In this case, older people, together with those with higher education and income levels, were included in the group of whom have a wider range of/engage more with disaster information repertoires. In particular, elderly people seemed to prepare themselves better than younger people. On the other hand, higher the level of user sophistication, higher the source of disparity [38]. In fact, SM are not designed in such a way to enable the use by people with a wide range of different disabilities [43]. Samuels and Taylor 2010 [44] have discussed that highly damaged areas with more elderly people, disabled people, and people without access to vehicles have instead a significant negative correlation with Twitter activity during disaster.

Nevertheless, Kent and Capello 2013 [64] demonstrated that young people (age under 18) may be a resource during disasters, and that more valuable disaster-related information is generated in those places that are characterized by more young people, denser population, and higher awareness levels. Similarly, Xiao et al. 2015 [106] revealed that communities with younger, male, and educated people were likely to use Twitter during Hurricane Sandy. In literature many other works can be found that confirm the same point of view, like Li et al. 2013 [107] who analyzed the use of Twitter and Flickr in order to identify the socioeconomic and demographic factors of the users. They showed that well-educated people working in business, management, science, and arts, such as young and urban dwellers, were more likely to use social media than the older and rural population. However, these works lack to tell whether these differences mainly affect people with lower status [108]. The sharing of information and the willingness to discuss facilitate the social construction of risk and can help people to develop individual skills useful to cope with disasters.

To conclude, adopting only the geographical perspective, Lai et al. 2018 [37] have revealed that citizens in developed countries are more active in using advanced media technologies in disasters, such as internet, social media, and mobile technology.

## 6. Conclusions

As discussed in this paper, DRP and vulnerability are two key concepts related to resilience and frequently discussed in the disaster literature. Accordingly, a literature review has been provided on the two concepts in the context of the digital space, with the aim of understanding how social virtual platforms can interact with and affect DRP and vulnerability. This work raises by the acknowledgment that a discussion that considers the relation between both the concepts and their role in the use and usefulness of SM in the DRM is lacking today and needs to be further implemented.

As emerged in the literature review here provided, there are a lot of potential positive reasons to use SM in circulating and collecting information in disasters, with potential positive effects on risk perception and vulnerability. However, a lot of challenges have been similarly identified that could negatively affect risk perception and vulnerability.

Thus, the study has been focused on understanding what are the main factors considered in the analyzed works as impacting on the usefulness of SM in disasters and what are the potentialities and challenges associated with their use.

On the basis of the analysis here provided, two roles for SM in DRM appear as potentially relevant: one is the role that virtual space can have in "reducing distances" and giving support to people in a difficulty; and the other is the role that SM platform can have in giving voice to groups that usually do not receive attention/space in the decision-making process. In particular, the role of SM can be summarized by the word "connectivity", here interpreted as the "connection" or "link" among individuals, mediated by a technological support/device, to create new virtual/real social relations and networks and promote transformation of the system. Furthermore, several studies show that SM could return a more dynamic image of the local situation compared to the traditional emergency planning (e.g., supply distribution), that are usually based on census data and where the data provide static information that does not necessarily correspond to the new scenario that disaster has created.

However, on the other hand several challenges have been identified in the use of SM in disasters. First of all, focusing on vulnerability, SM use could increase social and geographical disparities in those places exposed to risks and consequently on the people's capacity to prepare and respond to hazards. This is due to the circumscribed representativeness of the population in such platforms. Those who have no access to technology are automatically excluded and, therefore, most of the analyses based on SM data risk to not give a realistic representation of vulnerability and to increase informational vulnerability.

Accordingly, accessibility seems a fundamental entry point to ensure the spreading of quality of information and the effectiveness of social media communication, reducing the risk of informational vulnerability, and potentially increasing disaster risk perception. Accessibility, considered at different levels, could ensure not only the possibility to receive information but also to understand and use it in the right way. This could have effects also on the level of trust in the communication system.

On the other hand, about risk perception, the main challenges are connected to the reliability of information and the trust in the data themselves and in their sources. This is usually an obstacle in the use of such platforms in DRM and especially in the emergency phase. Furthermore, trust and experience are recognized as independent variables that could affect in different ways the communication flow. Accordingly, actions to reduce their impact should be seriously taken into consideration, acknowledging however that their effects can only be reduced and not totally eliminated.

Consequently, the study of disaster risk perception and vulnerability together has shown the strong interconnections that exist between the two fields of research and the need to move towards an implementation and integration of the two fields. Especially focusing on informational vulnerability, it can be observed how this is highly impacted by factors that usually are investigated under disaster risk perception studies. Yet, what has been observed is that although, e.g., DRP papers do not focus directly on vulnerability, frequently they take in consideration aspects that can be brought directly to it and vice versa (see for example the frequent presence of accessibility in studies about DRP, and of trust in studies about vulnerability). However, this has potentially negative consequences on the implementation of knowledge about the topic, as both the research fields continue to move independently, without benefiting from the knowledge produced in the other field. This is furtherly showing the increasing need for studies of this kind.

Furthermore, another limit identified among the investigated works has been recognized in the lack of a specific focus on vulnerable people when assessing and evaluating the use of SM in DRP. This aspect should be better addressed in future works on the topic, such as the limits identified in the geographic area covered by the studies. Works focusing on U.S. territory are numerically in preponderance, and geographically wider contributions are needed for the strategic sustainability of future actions in a wider variety of social contexts.

To conclude, more studies on the topic are expected for the future with the purpose to understand how to overcome the obstacles identified and to produce a useful interaction between social media communication and traditional one.

**Author Contributions:** Conceptualization, S.M., V.P. and S.B.; methodology, S.M., V.P. and S.B.; formal analysis, V.P.; investigation, V.P., O.N. and S.B.; data curation, S.M., V.P., O.N. and S.B.; writing—original draft preparation, S.M., V.P. and S.B.; writing—review and editing, S.M., V.P. and S.B.; visualization, V.P.; supervision, S.B. All authors have read and agreed to the published version of the manuscript.

**Funding:** The paper is result of the LINKS (Strengthening links between technologies and society for European disaster resilience) project that has received funding from the European Union's Horizon 2020 Research and Innovation Programme under Grant Agreement No. 883490.

**Institutional Review Board Statement:** Not applicable.

**Informed Consent Statement:** Not applicable.

**Conflicts of Interest:** The authors declare no conflict of interest. The funders had no role in the design of the study; in the collection, analyses, or interpretation of data; in the writing of the manuscript, or in the decision of publishing the results.

## Appendix A

**Table A1.** List of paper considered in this review work and focused on DRP. For each work (column 1) are highlighted: the considered hazard/s (column 2), the used/analyzed technological service/s (column 3), the involved stakeholders (column 4), and the main field, among those conceptualized in Section 3.2, in which the use of SM primary emerges (column 5). The papers with * have been identified using snowball in papers discussing DRP.

| Analyzed Paper | Kind of Hazard | Technological Services | Stakeholders | Primary Concept/s in Relation to DRP | Secondary Concept/s in Relation to DRP |
|---|---|---|---|---|---|
| Vieweg et al., 2010 [61] * | Flood and grassfire (multi-hazard) | Microblog (Twitter) | People "who were on the ground" during the event | - Information/communication flows<br>- Individual aspects | - Awareness |
| Alexander, 2014 [62] | General overview (Floods, earthquakes, tsunamis, and hurricane) | Social media (Twitter, and Facebook) | Review-based analysis | - Information/communication flows | - Quality information and reliability<br>- Social, demographic, and geographic differences |
| Chatfield et al., 2014 [76] | Eruptions | Social media (Twitter) | IT (e.g., e-government websites) | - Trust<br>- Information/communication flows | - Trust<br>- Quality information and reliability |
| Cheng et al., 2016 [103] | Earthquake | Social media (Facebook, Twitter, and YouTube) | 2047 Internet surveyed people | - Individual aspects | - Experience |
| Reuter et al., 2016 [55] * | Floods, heavy rain, wildfires, freezing rain, and storm (multi-hazard) | Social media (Facebook, Twitter, YouTube, and WhatsApp) | 761 emergency service staff | - Social aspects<br>- Individual aspects | - Social, demographic, and geographic differences<br>- Quality information and reliability<br>- Experiences |

**Table A1.** *Cont.*

| Analyzed Paper | Kind of Hazard | Technological Services | Stakeholders | Primary Concept/s in Relation to DRP | Secondary Concept/s in Relation to DRP |
|---|---|---|---|---|---|
| Silver & Matthews, 2016 [69] | Tornado | Social media (Facebook) | Residents | - Information/communication flows | - Quality information and reliability<br>- Social, demographic, and geographic differences |
| Mehta et al., 2017 [56] | General overview (cyclones, storm, floods, earthquake, fire, and hurricanes) | Social media (Facebook and Twitter) (as model for online trust in disasters within social media)—interrelation | Review-based analysis | - Trust | - Trust |
| Jurgens & Helsloot, 2018 [67] | General overview (floods, forest fires, earthquakes, and hurricane) | Social media (Facebook and Twitter) | Review-based analysis | - Information/communication flows | - Quality information and reliability<br>- Awareness |
| Reuter & Kaufhold, 2018 [16] * | General overview (floods, fires, volcanic eruption and related events—*Lahar, Floods and Debris Flows*—earthquakes, hurricanes, landslide, tornado, cyclones, hurricane, and typhoon) | Social media (Facebook and Twitter) and general discussion on interactions (Citizens to Citizens; Authorities to Citizens; Citizens to Authorities; Authorities to authorities) | Review-based analysis | - Information/communication flows | - Quality information and reliability |
| Bec and Becken 2019 [73] | Cyclones | Social media (Twitter and Facebook) | Twitter data analysis | - Social aspects<br>- Individual aspects | - Awareness<br>- Experience |
| Kaufhold et al., 2019 [70] | Floods and earthquakes (multi-hazard) | Social media (Facebook and Twitter) | Adults (1024 participants) | - Individual aspects | - Quality information and reliability<br>- Awareness |
| Reuter et al., 2019 [58] | Floods, earthquakes, and thunderstorms (multi-hazard) | Social media (General overview, most cited: Facebook and Twitter) | 7071 citizens | - Social aspects<br>- Individual aspects<br>- Trust | - Quality information and reliability<br>- Social, demographic, and geographic differences |
| Walkling and Haworth, 2020 [54] * | Flood | Information technologies (e.g., social networks) | Retired older adults | - Social aspects | - Social, demographic, and geographic differences |
| Geng et al., 2021 [68] | Floods | New Media | Weibo data analysis | - Social aspects<br>- Information/communication flow<br>- Trust | - Quality information and reliability<br>- Trust<br>- Awareness |
| Mohanty et al., 2021 [97] | Hurricane | Social media (Twitter) | 16,598 Twitter users (Twitter community in Florida, USA) | - Information/communication flow | - Quality information and reliability |

**Table A1.** *Cont*.

| Analyzed Paper | Kind of Hazard | Technological Services | Stakeholders | Primary Concept/s in Relation to DRP | Secondary Concept/s in Relation to DRP |
|---|---|---|---|---|---|
| Weyrich et al., 2021 [96] * | Floods | Social media (Twitter) | practitioners and PhD students involved in disaster risk management in various countries worldwide (20 players) | - Information/communication flow<br>- Trust | - Quality information and reliability<br>- Trust |
| Wu et al., 2021 [74] | Typhoon | Social media (Twitter) | Twitter data analysis | - Social aspects | - Quality information and reliability<br>- Awareness |
| Zhuang et al., 2021 [95] | COVID-19 | Social media (WeChat) | Online survey | - Social aspects<br>- Trust | - Trust<br>- Awareness<br>- Quality information and reliability |
| Hassan et al., 2022 [98] | COVID-19 | Social media (Twitter, Facebook, Instagram, YouTube, and WhatsApp) | 512 students and academics | - Trust | - Trust<br>- Awareness |

**Table A2.** List of paper considered in this review work and focused on vulnerability. For each work (column 1) are highlighted: the considered hazard/s (column 2), the used/analyzed technological service/s (column 3), the involved stakeholders (column 4), and the main field, among those conceptualized in Section 3.2, in which the use of SM primary emerges (column 5). The papers with * have been identified using snowball in papers discussing vulnerability.

| Analyzed Paper | Kind of Hazard | Involved Technology | Analyzed Stakeholders | Primary Concept/s in Relation to Vulnerability | Secondary Concept/s in Relation to Vulnerability |
|---|---|---|---|---|---|
| Shklovski et al., 2010 [89] * | Hurricane | Information and communications technology | Musicians in New Orleans, USA (40 interviews) | - Information/communication flow | - Quality information and reliability<br>- Experience<br>- Accessibility |
| Earle et al., 2011 [75] * | Earthquake | Social media (Twitter) | Twitter users | - Information/communication flow | - Quality information and reliability<br>- Awareness |
| Velev and Zlateva 2012 [63] | | | Review-based analysis | - Individual aspects<br>- Information/communication flow | - Awareness |
| Chatfield and Brajawidagda 2013 [109] | Tsunami | Social media (Twitter, Facebook, YouTube) | Twitter data analysis | - Information/communication flow | - Quality information and reliability<br>- Awareness |
| Kent and Capello 2013 [64] * | Fire | Social media (Multiple broadcasts that are likely to produce crowdsourced content: Instagram, Twitter, Flickr, and Picasa) | Social networks users | - Social aspects<br>- Individual aspects | - Social, demographic, and geographic differences<br>- Awareness |

**Table A2.** *Cont.*

| Analyzed Paper | Kind of Hazard | Involved Technology | Analyzed Stakeholders | Primary Concept/s in Relation to Vulnerability | Secondary Concept/s in Relation to Vulnerability |
|---|---|---|---|---|---|
| Schmeltz et al., 2013 [88] * | Hurricane (Sandy) | Social media (Facebook and Twitter) | Non-profit organization in Brooklyn, NY | - Information/communication flow<br>- Social aspects | - Social, demographic, and geographic differences |
| Fadaee and Schindler 2014 [87] | Hurricane (Sandy) | Social media | Occupy movement (New York) | - Social aspects | - Social, demographic, and geographic differences |
| Kongthon et al., 2014 [77] * | Flood | Social media (Twitter) | Twitter community (Thai people) | - Information/communication flow | - Quality information and reliability<br>- Awareness |
| Kent and Ellis, 2015 [43] | Natural hazards: generic panorama | Social media (YouTube, Facebook, Blogs, Twitter, Instagram, LinkedIn, MySpace, Flickr, and Google+) | Review-based analysis (People with disability) | - Information/communication flow<br>- Social aspects | - Social, demographic, and geographic differences<br>- Accessibility |
| Imran et al., 2015 [83] * | Natural hazards: generic panorama | Social media (Twitter) | Review-based analysis | - Information/communication flow | - Awareness |
| Madianou 2015 [41] * | Typhoon | Social media (General overview about all communicative opportunities not specified) and mobile media (e.g., sms phone) | multi-sited ethnography: local communities (101 participants) and 38 experts (representatives from humanitarian organizations, other civil society groups, government agencies, telecommunications companies, and other digital platform developers), while retaining a social class and gender balance | - Individual aspects<br>- Social aspects | - Social, demographic, and geographic differences |
| Xiao et al., 2015 [106] * | Hurricane | Social media (Twitter) | Twitter community (Aggregation of individuals in a certain geographic area) | - Social aspects | - Social, demographic, and geographic differences<br>- Awareness<br>- accessibility |
| McCallum et al., 2016 [14] | Floods | Social media (Twitter) | Review-based analysis | - Information/communication flows<br>- Social aspects | - Social, demographic, and geographic differences |

**Table A2.** *Cont*.

| Analyzed Paper | Kind of Hazard | Involved Technology | Analyzed Stakeholders | Primary Concept/s in Relation to Vulnerability | Secondary Concept/s in Relation to Vulnerability |
|---|---|---|---|---|---|
| Veer et al., 2016 [65] | Earthquake | Social media (Twitter, Facebook, Quakestories, and Stuff Earthquake Map) | local community: residents | - Information/communication flows<br>- Social aspects | - Experience<br>- Awareness |
| Checker, 2017 [86] | Storm and flood (Multi-hazard: connected events) | Social media (Twitter and Facebook) | Activists "Stop FEMA now" movement (a coalition of U.S. flood disaster survivors and other coastal homeowners) | - Social aspects | - Social, demographic, and geographic differences |
| Howard et al., 2017 [59] | Multi-hazards | Social media (Twitter and Facebook) | five potentially vulnerable groups in three key localities | - Social aspects<br>- Trust<br>- Information/communication flow | - Experience<br>- Quality information and reliability<br>- Trust<br>- Social, demographic, and geographic differences |
| Martín et al., 2017 [90] * | Hurricane | Social media (Twitter) | Inhabitants affected by the evacuation issue | - Social aspects | - Awareness<br>- Experience |
| Moorthy et al., 2018 [60] | Multi-hazard | Social media (Twitter and Facebook) | Review-based analysis | - Trust<br>- Information/communication flow | - Quality information and reliability<br>- Trust |
| Zhang et al., 2018 [79] | Flood | Social media (e.g., Twitter) | | - Information/communication flow | - Quality information and reliability |
| Zou et al., 2018 [40] * | Hurricane | Social media (Twitter) | Twitter community in 126 U.S counties affected by Hurricane Sandy | - Information/communication flow<br>- Social aspects | - Social, demographic, and geographic differences |
| Bhavaraju et al., 2019 [84] | Tornadoes, winter storms, wildfires, and floods (multi-hazard) | Social media (Twitter) | U.S. Twitter community | - Social aspects | - Social, demographic, and geographic differences |
| Harrison and Johnson 2019 [94] * | Natural hazards: generic panorama | Social media (Twitter, Facebook, YouTube, Periscope, Vine, Instagram, and Flickr) | Interviews to 15 government officials from 14 Canadian agencies | - Trust | - Trust |
| Nicholson et al., 2019 [17] | Hurricane | Social media (Twitter and Facebook) | Extended area community (Harris County, Texas, USA) | - Social aspects | - Accessibility<br>- Social, demographic, and geographic differences<br>- Experience |

**Table A2.** *Cont.*

| Analyzed Paper | Kind of Hazard | Involved Technology | Analyzed Stakeholders | Primary Concept/s in Relation to Vulnerability | Secondary Concept/s in Relation to Vulnerability |
|---|---|---|---|---|---|
| Wang et al., 2019 [39] | Hurricane | Social media (Twitter) | Vulnerable communities (physically and socially) | - Social aspects | - Social, demographic, and geographic differences |
| Wu et al., 2019 [81] | Floods | Social media (WeChat) | Text data analysis | - Social aspects | - Awareness |
| Wu et al., 2019 [110] | Typhoons | Social media | Text data analysis | - Social aspects | - Awareness |
| Fan et al., 2020 [85] | Hurricane | Social media (Twitter) | local community (From super-neighborhood scale to city scale) | - Social aspects<br>- Information/communication flow | - Quality information and reliability<br>- Social, demographic, and geographic differences |
| Martín et al., 2020 [66] | Hurricane | Social media (Twitter) | displaced/migrated residents and incoming tourists | - Social aspects | - Accessibility<br>- Experience |
| Sarker et al., 2020 [91] | | Social media (Twitter, Facebook, WhatsApp, and WeChat) | Review-based analysis | - Information/communication flow | - Social, demographic, and geographic differences<br>- Awareness |
| Wu et al., 2020 [82] | Floods | Social media (WeChat) | Text data analysis | - Social aspects | - Awareness |
| Chen and Ji 2021 [80] | Hurricane | Social media (Twitter) | Text data analysis | - Social aspects | - Quality information and reliability |
| Zhang et al., 2021 [78] | Hurricane | Social media (Twitter) | Emerging influential contributors text data analysis | - Social aspects<br>- Information/communication flow | - Quality information and reliability<br>- Social, demographic, and geographic differences<br>- Awareness |

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
