# Peer review of "Framing Disaster Risk Perception and Vulnerability in Social Media Communication: A Literature Review"

_sustainability, doi:10.3390/su14159148_

Round 1
Reviewer 1 Report
See the attached file.

Reviewer 2 Report
The paper represents the strict and correct analysis of the vulnerability of populations and of the disaster risk management, with various components of the disaster behavior regulation through awareness. The social media are considered as a tool to enhance the informational resources and to reduce the disaster risk for population with improving its perception.
The trainings and protocols represent the traditional way to cope with a disaster, especially, sudden and serious one, the social media as communication tool can help to inform and to prepare people to ask questions about potential first steps and behavior in case of.
The impact of the awareness through social media on the disaster risk perception and readiness to face it, is analysed through the knowledge base construction on the basis of EU Links project combined with the study of 58 scientific documents' presented in Scopus, WoS and Google Scholar.
The outcomes and discussions are of high importance and create a clear basis for the further empirical research, especially, to verify the hypotheses about the positive or negative influence of the reliability and trust to the disaster management.
There are several questions:
Regarding number of papers studied:
the text of the paper mentions 58 selected papers, but further 49 ones are mentioned and another 12 were added. Now many papers were examined finally? 58, 49 or 61?
- line 334 - "99% of papers" - in case of 58 analysed papers, 99% is equal to 57.5 papers. May be, 98%?
About the results:
- lines 346-347 - "classified under the “World” category (purple in Figure 3)" - the Fig.3 contains regions, where there no "World" category.
Some typos are to be checked:
- line 85 - "to improve citizen understanding of risksand to improve" - probably, " ... risk and ..."
- line 103 - "the five step of the approach" - "five steps"?
- line 189 - "It entailsed the..."
These minor remarks do not reduce the great interest of the research.
Hope to see the paper published soon.
Reviewer 3 Report
The authors report “the results of a literature review on how social media can impact on disaster risk perception and vulnerability and how these two aspects are interconnected”.
Disaster management and the role of social media in it, both are important topics including risk perception and vulnerability.
The manuscript could be useful; however, it does not clearly present and establish its case. A proper literature review that could help us understand its case and contributions compared to the existing literature is not available. Some references to consider:
https://www.mdpi.com/2071-1050/14/2/810
https://geoenvironmental-disasters.springeropen.com/articles/10.1186/s40677-021-00181-3
For these reasons, it is not clear what is needed, what is being done here, how it is being done here, what are the results and findings in relation to the stated aim and objectives, and whether they are consistent with the methodology and the presented/extracted evidence.
Let me explain further.
The problem statement and the contributions are not clear. What is the exact problem this literature review is trying to address, what is the methodology to do so, what are the findings, and how do these findings bridge the existing gaps in the knowledge, theory, and practice?
For example, the material from the second last paragraph of the introduction section could go earlier in the section to improve the readability. The point is that the authors should think carefully about the main research gap they are trying to address and clearly outline the research gap as well as a strong explanation and reference support for that research gap. For this purpose, all the component topics or concepts (vulnerability, perception, resilience, Disaster Risk Management (DRM), Disaster Risk Perception (DRP), Disaster Risk Reduction (DRR)) should be briefly and clearly introduced particularly when those concepts are not clearly established in the literature with clearly defined agreed-upon reasons, i.e., there are latitude and vagueness in their definitions and treatment of the concepts in the literature. This should follow by a clear statement about the aim and focus of the research, e.g., the interconnectedness of risk perception and vulnerability in disaster research. The authors then should clearly brief the methodology to achieve their proposed work; such as what is the methodology of their research and how it achieves the proposed aim and objectives (e.g., how does the proposed methodology in the paper help us establish that “social media is opening issues about how risk perception and vulnerability are potentially interconnected and how this relationship is also shaping different ways of conceptualizing informational vulnerability”. How does the methodology used in this manuscript help address the stated gaps in the current research and practice? Of course, all these statements should be brief and clear with pointers to the details in the later sections.
The issue of social approach lacking in emergency response or disaster management systems should be discussed from the beginning in detail and supported by strong references. The following statement is only supported by a reference from 1991 so how does it hold today? Should be clarified. “Despite these calls to a social approach, for years the emergency response system has favoured the technological and physical sciences, neglecting the contribution that the social sciences could give in the field of Disaster Risk Perception (DRP) and Reduction (DRR) [8_Alexander, 1991].”. The subsequent paragraph is not supported by appropriate references such as “…importance of a social perspective in DRM has 38 been introduced in the Hyogo Framework 2005-2015 and then in the Sendai Framework 39 for Disaster Risk Reduction 2015-2030…”.
The issue raised in the paragraph above is the core point in the case for your manuscript and therefore must be established strongly and with appropriate references.
The authors should briefly define the scope of the term “disaster”. What are a disaster and its scope that you consider in this manuscript? The use of the terms DRR, DRM, and DRP in the manuscript is not clear. Clarity in this respect is needed particularly because these terms form the main case and aim of the work.
Writing clarity and proofreading are required throughout the manuscript. Example in the abstract, Line 13-14: “and eventually how.”. Line 85: risksand. Line 189: entailed.
The reference style does not look like the one used by MDPI.
Round 2
Reviewer 1 Report
Thank you for the changes, the new MS is much improved. Marking as "accept after minor revisions" because there is still quite a bit of polishing to do on the English.
Reviewer 3 Report
The manuscript has been improved. I see that the authors did not address most of my comments. I understand that the manuscript makes some contributions to the literature however these have not been properly communicated. Addressing my concerns would have improved the readability and motivation for the work.